# Influence of Nutritional Intakes in Japan and the United States on COVID-19 Infection

**DOI:** 10.3390/nu14030633

**Published:** 2022-02-01

**Authors:** Yasuo Kagawa

**Affiliations:** Department of Medical Chemistry, Kagawa Nutrition University, Saitama 350-0288, Japan; kagawa@eiyo.ac.jp; Tel.: +81-492-82-3618

**Keywords:** COVID-19, obesity, saturated fat, EPA/DHA, soybean, diabetes, mortality, Japanese

## Abstract

The U.S. and Japan are both democratic industrialized societies, but the numbers of COVID-19 cases and deaths per million people in the U.S. (including Japanese Americans) are 12.1-times and 17.4-times higher, respectively, than those in Japan. The aim of this study was to investigate the effects of diet on preventing COVID-19 infection. An analysis of dietary intake and the prevalence of obesity in the populations of both countries was performed, and their effects on COVID-19 infection were examined. Approximately 1.5-times more saturated fat and less eicosapentaenoic acid/docosahexaenoic acid are consumed in the U.S. than in Japan. Compared with food intakes in Japan (100%), those in the U.S. were as follows: beef 396%, sugar and sweeteners 235%, fish 44.3%, rice 11.5%, soybeans 0.5%, and tea 54.7%. The last four of these foods contain functional substances that prevent COVID-19. The prevalence of obesity is 7.4- and 10-times greater in the U.S. than in Japan for males and females, respectively. Mendelian randomization established a causal relationship between obesity and COVID-19 infection. Large differences in nutrient intakes and the prevalence of obesity, but not racial differences, may be partly responsible for differences in the incidence and mortality of COVID-19 between the U.S. and Japan.

## 1. Introduction

The coronavirus disease 2019 (COVID-19) has become a global pandemic. Key questions are why certain people are more susceptible to COVID-19 and the roles of good nutrition to support immune function. Japan and the U.S. are modern democratic countries with populations of over 100 million. Although both countries have in common their loose COVID-19 regulations, the numbers of patients and deaths per million people due to COVID-19 differ markedly (119,026 and 1965, respectively, in the U.S. compared with 11,381 and 126, respectively, in Japan, as of 28 August 2021) [1]. Even in the vaccinated, these differences are very large: cases and deaths per million were 167,302 and 2537, respectively, in the U.S. but only 13,776 and 146, respectively, in Japan, as of 2 January 2022 [1]. Therefore, the present study focused on the reasons for these differences. Many factors have been investigated to elucidate the causes of ethnic disparities in COVID-19 cases and deaths in the U.S. and Japan [2]. These factors include the effects of diet and the role of the immune system, which is highly affected by nutritional deficits and obesity, leading to decreased immune responses and a consequent higher risk of infection and disease severity [3]. A systematic literature review has identified that good nutrition can help prevent COVID-19 [3]. Regarding socioeconomic factors, the greatest difference between the countries is the proportion of the elderly (≥65 years), which is twice as high in Japan as in the U.S. (28.1% vs. 15.8%) [4]. However, the minority populations of Blacks and Hispanics in the U.S. are also vulnerable to COVID-19 [2]. Thus, it is difficult to explain differences in vulnerability to COVID-19 between Japan and the U.S. in terms of socioeconomic factors alone. Obesity makes the host highly vulnerable to COVID-19 by decreasing immunological resistance [5]. A search for “COVID-19 and mortality and obesity” in PubMed found 1209 papers, and an additional search for “diet” found 39 papers. However, when “Japanese diet” was used as the search term, no papers were found.

For the purpose of this study, the hypothesis was that the dietary difference between Japan and the U.S. is one of the major factors explaining the large difference in COVID infection between the two countries. Thus, the aim of the present study was to examine the effects of nutritional factors, including the frequency of obesity, on vulnerability to COVID-19 in Japan.

## 2. Materials and Methods

PubMed was used as the major search engine. In addition, other search engines, including CiNii (Japanese database), were also used, with the following keywords: COVID-19; obesity; saturated fat; EPA/DHA; soybean; diabetes; mortality; and Japanese. The 19 papers found by searching for COVID-19 and Japanese diet in PubMed were irrelevant to the subject of this paper. Differences between Japan and the U.S. were compared in terms of the following COVID-19 pandemic data [1,2]: nutritional prevention [3], population [4], obesity and immunity [5], non-communicable diseases [6], ethnic differences in obesity [7], ethnic disparity of viral infection [8], diabetes mellitus [9,10], genetic polymorphism [11], impact of westernization [12], and nutritional intakes in the U.S. [13] and Japan [14]. The inclusion and exclusion criteria of the references were a date after the COVID-19 pandemic in the case of health surveys, and the nutrient and food intakes were the most recent nationwide data available: United States Department of Agriculture NHANES 2017–2018 [13] and The National Health and Nutrition Survey Japan, 2019 [14]. The author has accumulated data on nutrition and health surveys in Japan and the U.S. for the past 44 years [12], but since COVID-19 is a pandemic that started after 2020, Forest plots of many different past and local data were not performed. Unfortunately, after the COVID-19 pandemic, these nationwide health–nutrition surveys were stopped. The sample sizes of NHANES 2017–2018 (ages 40–49 years) [13] were 340 and 367 for males and females, respectively, and those of the National Health and Nutrition Survey Japan 2019 [14] were 363 and 353, respectively. Differences in intake between Japan and the U.S. were compared, showing significant differences in nutrients and foods. For example, the intake of saturated fatty acids in Japan was 20.3 ± 9.9 g for men and 20.0 ± 8.7 g for women, and the intake in the United States was 34.1 ± 1.6 g for men and 26.1 ± 1.4 g for women, with a highly significant *p*-value of 0.000. Reliable data on health, nutrient intakes, and food intakes are compared in Figure 1, Figure 2, Figure 3, Figure 4, Figure 5, Figure 6, Figure 7, Figure 8, Figure 9 and Figure 10. Daily intakes of the nutrients and foods of Japanese men and women were set to 100% and compared with the intakes of American men and women. Since the frequency of gene polymorphisms differs greatly between Japanese and Americans [11], Japanese Americans were used as an important control group. The relevant references are quoted according to each subject.

## 3. Results

### 3.1. Differences in Cases and Deaths between Japanese and U.S. Groups

The numbers of cases and deaths per million people in the U.S. were 12.1 times (167,302 vs. 13,776) and 17.4 times (2537 vs. 146) higher than in Japan, respectively, on 2 January 2022 [1]. On this date, the average numbers of cases and deaths per million people globally were 37,174 and 700, respectively [1].

In the U.S., COVID-19 morbidity (Figure 1) and mortality (Figure 2) are high in Black and Hispanic populations [2]. The ethnicities of the 65,143 adults hospitalized in the U.S. between 1 March and 24 October 2020 [2] are shown in Figure 1, and mortality according to ethnicity in the same period [2] is summarized in Figure 2. The COVID-19 vaccine was not available in Japan or the U.S. at that time. Similar numbers of cases (Figure 1) and deaths (Figure 2) observed in Asians living in the U.S. and in Whites [2] indicate that genetic differences between these ethnicities are unlikely to be the cause of the large differences in COVID-19 data between Japan and the U.S. Disease severity and mortality rates were significantly higher in elderly patients; however, the total population aged ≥65 years is 28.1% in Japan and 15.8% in the U.S. [4]. Therefore, the differences in COVID-19 cases and deaths between Japan and the U.S. cannot be explained solely by the proportions of the elderly population.

### 3.2. Prevalence of Obesity and Noncommunicable Diseases in Japan and the U.S.

Obesity is known as a risk factor for COVID-19. Several studies have shown that obese people are vulnerable to COVID-19 infection and death [5]. Figure 3 shows the prevalence of three risk factors for noncommunicable diseases in Japan and the U.S. [6]. The prevalence of obesity (body mass index: BMI > 30 kg/m^2^) is 7.4-times greater in U.S. males (35.5%) and 10-times greater in U.S. females (37.0%) than in Japanese males and females [6]. The incidence of obesity has increased steadily in the U.S. over the past 25 years. Smoking is more prevalent in the Japanese than in Americans [6]. The prevalence of hypertension (systolic blood pressure ≥ 140 mmHg or diastolic blood pressure ≥ 90 mmHg) in Japan is slightly higher in males (1.47 times) and females (1.2 times) compared with the U.S. (Figure 3) [6]. Obesity leads to chronic inflammation and lowered immunity [5]. Obesity was found to be associated with a higher intubation/mechanical ventilation rate and increased mortality in COVID-19 patients (*p* = 0.002) [15]. Moreover, obesity is a known risk factor for cardiovascular diseases, hypertension, type 2 diabetes mellitus, respiratory diseases, and some types of cancer [5].

The rarity of obesity among Japanese people living in Japan is not fully explained by genetic differences compared with Whites. The prevalence of being overweight in Asians living in the U.S. was higher among all Asian subgroups compared with white men, except Filipinos [7]. The prevalence ratio of obesity in Asian men other than Chinese, Filipinos, and Indians compared with white men was 0.96 (95% CI: 0.78–1.18) [7]. Obesity and type 2 diabetes mellitus, which are risk factors for COVID-19, may underlie the health disparities among these groups. Minorities have difficulties in accessing healthy food, nutritional education, and healthcare in the U.S. [8].

The prevalence of type 2 diabetes mellitus is higher among Asians living in the U.S. [9] than among Japanese people living in Japan [10], and this may contribute to the difference in susceptibility to COVID-19 between these countries. The weighted age- and sex-adjusted prevalences of total diabetes were 12.1% (95% CI: 11.0–13.4%) for White, 20.4% (95% CI: 18.8–22.1%) for Black, 22.1% (95% CI: 19.6–24.7%) for Hispanic, and 19.1% (95% CI: 16.0–22.1%) for Asian adults (overall *p* < 0.001) [9]. On the other hand, the prevalence of diabetes mellitus in Japan remained largely unchanged over the years (1990: 7.9%; 2010: 7.9%) [10]. Some types of diabetes mellitus, including type 1 diabetes mellitus, mitochondrial diabetes [16], and metabolic disorders [17], are not related to dietary intake, but a genetic analysis showed a low incidence [16,17]. Hypertension was shown to be one of the risk factors for COVID-19 infection [18]. Nutrigenomic studies have shown that Japanese persons are more vulnerable to type 2 diabetes mellitus and hypertension because the frequencies of the risk genetic polymorphisms for these diseases are evolutionarily high to save energy and salt in rainy subtropical districts [11]. In fact, the prevalence of hypertension is higher in Japan than in the U.S. (Figure 3) [6]. Despite this genetic vulnerability, the Japanese diet partially prevents severe diabetes mellitus and hypertension [10,11].

### 3.3. Effects of Different Nutrient Intakes between Japan and the U.S. on COVID-19 Infection

The high prevalence in the U.S. of obesity and noncommunicable diseases, which are risk factors for COVID-19, may be caused by the typical U.S. diet, consisting of large amounts of saturated fat and sugars and low contents of fiber, n-3 polyunsaturated fats, and antioxidants. Although the impact of westernization on the nutrition of the Japanese people has changed the traditional diet over the last 44 years, there are still large differences in nutrient intakes between the U.S. and Japan [12].

Improved nutrition to support the immune system may be a viable approach to both prevent and alleviate COVID-19 infection [3]. These nutrient factors will be discussed in the following sections.

#### 3.3.1. Difference in Intakes of Saturated Fat, n-6 Polyunsaturated Fatty acids, and Sugar

Figure 4 shows the macronutrient intakes of males and females aged 40–49 years in the U.S. [13] relative to those in Japan [14], which are set as 100%. The intakes of fats and saturated fatty acids (stearic and palmitic acids) were higher in the U.S. than in Japan. The intakes of n-6 polyunsaturated fatty acids, including linoleic acid (LA) and arachidonic acid (AA), were also higher in the U.S. than in Japan (Figure 5) [13,14]. Saturated and n-6 polyunsaturated fatty acids are both associated with the development of obesity, type 2 diabetes mellitus, a pro-inflammatory profile, and atherosclerosis. Saturated fatty acids are linked with epigenetic changes [19]. Sugar is rapidly converted into saturated fatty acids. The intakes of sugars and sweeteners were 66.8 and 27.1 (kg/capita/year) in the U.S. and Japan, respectively [20]. Consumption of sugar-sweetened beverages has been shown to be associated with incident type 2 diabetes mellitus [21]. Thus, the large intake of saturated fats and sugar in the U.S. causes obesity and type 2 diabetes mellitus and may lead to increased risks of COVID-19 infection and mortality. The U.S. diet, rich in n-6 polyunsaturated fatty acids and low in n-3 polyunsaturated fatty acids together with a large sugar intake, causes obesity and low-grade inflammation [22]. The US diet contains large amounts of n-6 fatty acids (including arachidonic acid, which is converted into leukotrienes, and thromboxanes) and increased cytokines, such as interleukin (IL)-1, IL-6, and tumor necrosis factor, which are risk factors for COVID-19 [22]. Peripheral inflammation caused by COVID-19 may have long-term consequences in those that recover [22].

#### 3.3.2. Differences in Intakes of n-3 Polyunsaturated Fatty Acids

The intakes of eicosapentaenoic acid (EPA) and docosahexaenoic acid (DHA) are very low in the U.S. diet compared with those in the Japanese diet (Figure 5) [13,14]. High intakes of EPA and DHA from fish contribute to health and the prevention of cardiovascular diseases in Japanese islanders [23]. N-3 polyunsaturated fatty acids are mainly composed of α-linolenic acid (ALA), DHA, and EPA. Large amounts of dietary ALA can be converted into DHA through EPA by Δ5 fatty acid desaturase [24]; however, the efficiency of DHA synthesis is low, especially in carriers of the CC genotype of the *FADS1* gene (Δ5 fatty acid desaturase gene), and the C-allele is detected in about half of the Japanese population [24]. Both DHA and EPA are converted into protectin, which inhibits the hyperactivity of immune cells, and it also supports phagocytosis and neutrophil differentiation to prevent COVID-19 infection [22].

#### 3.3.3. Differences in Intakes of Vitamins between Japan and the U.S.

Figure 6 shows the comparison of vitamin intakes in the U.S. relative to those in Japan [13,14]. Intakes of vitamin D and K are lower in the U.S. than in Japan. Vitamins A, B complex, C, D, and E boost the immune system, and thus, deficiencies of these vitamins may predispose to COVID-19 infection [3]. Although the intakes of vitamin A, B complex, C, and D in Japan and the US do not meet the Japanese dietary intake standard [25] (e.g., the vitamin A standard of 850 μg retinol active equivalent [RAE] in males), an intake of 550 μg RAE may be sufficient in terms of immune system function. A detailed review of the entire vitamin B complex has shown its roles in the immune system and depression [26]. Intakes of polished rice and white bread in Japan and the U.S. contribute to the low intakes of vitamin B complex vitamins, but it is thought that, except in the low-income group, the degree of insufficiency is unlikely to weaken the immune system [26].

However, vitamin D has been recommended for the prevention of COVID-19. An intake of 250 μg/day has been shown to raise serum 25(OH)D concentrations above 40–60 ng/mL (100–150 nmol/L) [27]. The Japanese intake of vitamin D of 6.4 μg/day [14] is less than the standard of 8.5 μg/day for adults [25]. Intakes of vitamins D and K are both deficient in the U.S. (Figure 6) [13,14]. Serum 25(OH)D levels less than 12 ng/mL were found to be significantly associated with a higher risk of severe COVID-19 infection and death [28]. Low vitamin D_3_ leads to an increase in anti-inflammatory and immunoregulating interleukin 10 (IL-10) cytokines and reduces the frequency of Th17 cells, which, in turn, decreases IL-17 and proinflammatory cytokine tumor necrosis factor alpha (TNFα) production, decreasing inflammatory effects in the host [29]. Thus, vitamin D deficiency is often found in patients with COVID-19 [29]. Dark skin color that prevents vitamin D synthesis by ultraviolet irradiation has been reported to increase the risk of COVID infection in Blacks [29]. Observational studies have shown that serum 25-hydroxyvitamin D concentrations are inversely correlated with the incidence or severity of COVID-19 [29]. It is recommended that COVID-19 patients take vitamin K_2_ and magnesium, together with vitamin D, to avoid side-effects such as arteriosclerosis and osteoporosis [30]. In the Japanese diet, fish contains large amounts of vitamin D, and fermented soybean (Natto) (8.7 g/day) contains large amounts of vitamin K (600 μg/100 g) [14]. Thus, vitamin supplementation to support the immune system has been recognized as a viable approach to prevent COVID-19 infection [3]. In fact, vitamin fortification of cereals, including folic acid, reduces the incidence of many diseases and saves medical costs [31]. In a study of the effects of supplements, those taking n-3 polyunsaturated fatty acids, multivitamins, or vitamin D had a lower risk of COVID-19 infection by 12%, 13%, and 9%, respectively [32].

#### 3.3.4. Differences in Intakes of Minerals between Japan and the U.S.

Figure 7 shows the comparison of mineral intakes in the U.S. relative to those in Japan, which were set as 100% [13,14]. Calcium intake is higher in the U.S. than in Japan due to the higher intake of milk in the U.S. (Figure 7) [13,14]. Among macrominerals, sodium intake in both countries corresponded to about 10 g of salt per day, which is double the WHO recommendation of 5 g per day. The prevalence of hypertension, a risk factor for COVID-19, is approximately 1.5- and 1.2-times higher in males and females in Japan, respectively, than in the U.S. (Figure 3) [6]. This is due to the difference in the frequency of the risk T allele of angiotensinogen genes (*AGT*, rs699, T/M) between the Japanese (0.84) and Americans (0.41) [33].

The intakes of potassium in both Japan and the U.S. are approximately 60% of the Japanese recommendation (3 g) [25]. The major potassium sources are vegetables and fruits, intakes of which are lower than recommended, as has been described previously [14]. Low potassium intake causes hypokalemia, particularly in those infected with COVID-19. Hypokalemia was detected in 119/290 patients (41%) during hospitalization for COVID-19 (mean serum potassium: 3.1 ± 0.1 mEq/L) [34]. Hypokalemia is associated with hypocalcemia, which was detected in 50% of these subjects [34]. A low potassium level requires supplementation *per os* or by injection, and care should be taken on ECG assessment [34].

The intakes of magnesium in both Japan and the U.S. are slightly lower than the recommended Japanese values of 370 mg for males and 290 mg for females (Figure 7) [13,14]. Chronic magnesium deficiency can result in low-grade inflammation, which has been linked to predisposition to infectious diseases [35]. Magnesium is needed in the immune reaction as a cofactor for immunoglobulin synthesis and processes related to the functions of T and B cells [35].

Many micronutrients, including iron, zinc, selenium, and copper, have been shown to activate the immune system to prevent COVID-19 infection [3]. However, intakes of iron and zinc in both Japan and the U.S. do not meet the recommended standard [25]. Both zinc and copper are components of superoxide dismutase, which prevents the oxidative stress of inflammation [3]. Zinc is important for immunity mediators such as enzymes, thymic peptides, and cytokines [36]. Iron and selenium are components of catalase and peroxidase, respectively, which also prevent oxidative stress caused by COVID-19 infection [3].

### 3.4. Effects of Different Food Intakes between Japan and the U.S. on COVID-19 Infection

Nutrients are ingested in the form of food that contains bioactive substances in addition to the nutrients. Compared with the U.S., Japan consumes less meat (particularly beef), milk and dairy products, sugar and sweeteners, fruits, and potatoes, but more fish and seafood, rice, soybeans, and tea (Figure 8) [13]. Similar differences are seen in food intakes (Figure 9), with an extreme difference in seaweed intake [14,20]. These differences are reflected in the differences in nutrient intakes (Figure 4, Figure 5, Figure 6 and Figure 7), in the extreme difference in obesity prevalence (Figure 3), and perhaps also in the COVID-19 death and hospitalization rates (Figure 1 and Figure 2). A study that surveyed the effects of diet, obesity, and COVID-19 mortality in 188 countries showed that higher rates of consumption of meat, sugar and sweeteners, animal fats, and animal products were associated with more deaths in COVID-19 patients [37]. The consumption of sugar products had a considerable effect on mortality, and obesity was shown to increase death rates and reduce recovery rates from COVID-19 [37].

#### 3.4.1. Intakes of Red Meat and Milk

The dietary pattern of higher intakes of beef, milk, and dairy products (Figure 8) resulted in a large incorporation of saturated fatty acids and n-6 polyunsaturated fatty acids in the U.S. compared with Japan (Figure 4) [13,14]. Excessive dietary intake of saturated fatty acids is associated with increased risks of obesity, type 2 diabetes mellitus, and cardiovascular diseases, which are themselves risk factors for COVID-19 infection [38]. Integrating condensed milk into the diet increased weight gain and body fat formation [39]. A study of Western and Japanese dietary intakes recommended a saturated fat intake of approximately 20 g/day (approximately 10% of total energy), i.e., 200 g of milk/day and 150 g of meat every other day, to prevent cardiovascular diseases [38]. As shown in Figure 8 [20] and Figure 4 [13,14], these recommended intakes are very close to those consumed in Japan, but excessive intakes of these foods are consumed in the U.S. Of 428,070 UK Biobank participants, 100,175 (23.4%) were obese [40]. Red and processed meat consumption was associated with an increased risk of cardiovascular death (HR, 1.04; 95% CI: 1.01–1.08) per weekly serving for participants with obesity and 1.04 (1.02–1.07) for those without obesity after adjusting for age, sex, ethnicity, education, smoking and alcohol status, and overall health [40].

#### 3.4.2. Intake of Sugar and Sweeteners

The prevailing mechanism linking sugar-sweetened beverage intake to weight gain is decreased satiety and incomplete compensatory reduction in energy intake at subsequent meals following the consumption of liquid calories (Figure 4). On average, sugar-sweetened beverages contain 140–150 calories and 35–37.5 g of sugar per 340 g serving [41]. If the normal dietary intake does not decrease by an equivalent number of calories per serving, then weight gain is expected [41]. The short-term feeding of high-calorie sweetened beverages resulted in weight gain compared with non-caloric sweetened beverages [41].

#### 3.4.3. Intake of Fish

The dietary pattern of a high intake of fish (Figure 8 and Figure 9) [13,14,20] results in the high consumption of long chain n-3 polyunsaturated fatty acids such as EPA and DHA in Japan compared with the U.S. (Figure 5) [13,14]. Dietary intake of EPA/DHA is associated with a decreased risk of obesity and cardiovascular disease [23,42] because high consumers of fatty fish developed a favorable HDL-cholesterol-related lipoprotein profile and anti-inflammatory phenotype than low consumers of fatty fish [42]. The anti-inflammatory properties of EPA/DHA are caused by the production of protectin, which reduces morbidity and mortality from COVID-19 infection [42,43]. In the U.S., the odds ratio for the death of COVID-19 patients with a high omega-3 index (red blood cell EPA + DHA/total fatty acid ≥ total) compared with a lower omega-3 index was 0.25 (*p* = 0.07) [43]. In contrast, the omega-3 index of red blood cells of the average Japanese individual was 8.2, and even in those with a CC genotype of *FADS1*, it was 7.7 [44]. Thus, the higher intake of fish by the Japanese may be protective against death from COVID-19.

#### 3.4.4. Intake of Soybeans

Soybean intake is 200-times greater in the Japanese than in Americans (Figure 8) [20]. Soybeans contain isoflavones, which prevent obesity and inhibit SARS-CoV-2 in vitro [45]. The odds ratio for overall obesity was 0.91 (95% CI: 0.85, 0.98) in the highest versus the lowest soy isoflavone tertile group [45]. A negative association was also observed between soy isoflavone intake and central obesity [45]. The prevention of obesity by soy isoflavones has also been confirmed in a Chinese population [46]. Together with anti-viral effects [47], isoflavonoids are potential candidate inhibitors against the COVID-19 receptor ACE2 (angiotensin-converting enzyme 2) and COVID-19 protease [47]. Together with the prevention of obesity, soybean intake by the Japanese directly prevents COVID-19 infection.

#### 3.4.5. Intake of Seaweeds

Seaweeds are consumed by the Japanese but not eaten in the U.S. (Figure 9) [13,14]. Seaweeds contain peptides inhibiting angiotensin-converting enzyme 2, fucoidan (dietary fiber), EPA, fucoxanthin, vitamin D_3_, and phlorotannin [48]. These components exert antioxidant, anti-inflammatory, and antiviral effects directly, as well as indirectly, through prebiotic effects. The inhibitory components of angiotensin-converting enzyme 2 may prevent the entrance of the COVID-19 virus into the host cells [48]. Many algae-derived compounds that augment the immune response and mechanisms have been reviewed in detail [49].

#### 3.4.6. Intake of Green Tea

Approximately twice as much tea is consumed in Japan as in the U.S. (Figure 8) [20]. Green tea accounts for 68% (650 g/capita/year) of total tea consumed in Japan, but nearly 0% of that in the U.S. (personal communication from the International Tea Committee: Annual Bulletin of Statistics). Green tea, black tea, and epigallocatechin 3-gallate significantly reduced body fat compared with controls [50]. The effects of tea on improved glucose metabolism were studied in detail [51]. Green tea was more effective than black tea in reducing body fat and blood glucose of rats fed a 15% fat diet [50]. The greater effects of green tea compared with black tea are explained by its ~30% higher inductions of lipid metabolizing carnitine palmitoyl transferase-1 and acyl CoA oxidase [50]. The activation of Toll-like receptor 4 (TLR4) by saturated fatty acids is the cause of obesity-induced inflammation, but this is suppressed by epigallocatechin-3-gallate in green tea [52]. In a recent review, both green tea and black tea polyphenols exhibited antiviral activities against various viruses, especially single-stranded RNA viruses [53]. The higher intake of tea, especially the higher intake of green tea in Japan compared with the U.S., may partially explain the difference in mortality between the two countries.

#### 3.4.7. Intakes of Rice and Wheat

Although intakes of corn flour and corn meal and oats and oat flour are higher in the U.S. than in Japan, the absolute amount is very small (Figure 10) [13,14]. The intake of rice is seven-times greater in Japan than in the U.S. (Figure 10) [13,14]. Among G20 countries, the consumption of rice (per capita kg/year) is inversely proportional to the number of COVID-19 patients and deaths per million population [54]. Rice, especially unpolished rice, promotes a gut microbiome with highly prevalent healthy *Firmicutes* and a low prevalence of unhealthy *Fusobacterium* [54]. The phytonutrient profile of unpolished rice includes oligosaccharides, γ-oryzanol, and GABA, which may collectively enable this microbiome profile and function [54]. In contrast, a positive relationship was found between COVID-19 prevalence in G20 countries (and Spain) and wheat consumption [54]. This finding may partly reflect the increased usage of refined wheat flour in the U.S. [54]. Adipose tissue dysfunction was observed in a U.S. high-sucrose, high-fat diet group, which showed marked PPAR-γ underexpression and increased levels of cytokines and other inflammatory markers and oxidative stress [55]. In addition, γ-oryzanol prevented adipose tissue dysfunction and the cytokine storm in COVID-19 obese patients and promoted peroxisome proliferator-activated receptor γ overexpression [55].

### 3.5. Dietary Patterns

Since the diet is a collection of many foods and there are interactions between individual nutrients and foods, the differences in nutrient and food intakes between Japan and the U.S. alone cannot explain the large differences in the prevalence and mortality of COVID-19 between these countries. Thus, an international analysis of dietary patterns and mortality is important [56]. Evidence based on studies from 28 countries, including the U.S., Japan, and Australia, indicated consistent findings across the studies despite the variety of indices or scores used for the many different dietary patterns [56]. Of the Western diets, the Mediterranean diet is one of the healthy dietary patterns that improves the immune system and has been suggested to be useful in the prevention of COVID-19 [57]. Dietary patterns termed “nutrient-dense dietary patterns” in adults and older adults, which involved a higher consumption of vegetables, fruits, legumes, nuts, whole grains, unsaturated vegetable oils, fish, and lean meat or poultry, were associated with a decreased risk of all-cause mortality [56,57]. The intake of a plant-based diet was associated with 59% lower odds of moderate-to-severe COVID-19 severity [58] and a lower risk of COVID-19 infection [59]. The plant-based natural products used to cure patients infected with COVID-19 were reviewed [60]. Plants contain quercetin, and together with vitamin D and estradiol, they inhibit COVID-19 receptors, angiotensin-converting enzyme 2, etc. [61]. A healthy Japanese dietary pattern score similar to the Japanese standards [21] was significantly and positively correlated with the intakes of all 21 micronutrients in the standards [62]. These healthy patterns were also relatively low in red and processed meat, high-fat dairy, and sugar [56]. High-fat, high-sugar diet-induced changes in lipid metabolism are associated with increased COVID-19 severity and delayed recovery in the Syrian hamster [63]. The high rate of consumption of diets high in saturated fats and sugars in the U.S. may cause obesity and type 2 diabetes mellitus and increase risk for COVID-19 infection.

The lower mortality rate from COVID-19 in Japan is thought to reflect the lower prevalence of obesity in Japan than in the U.S., due to a diet low in red meat and saturated fatty acids and high in fish, n-3 polyunsaturated fatty acids, plant foods such as soybeans, and green tea (Figure 8 and Figure 9). The low mortality rate from ischemic heart disease was thought to reflect the low prevalence of obesity in Japan [64]. A low consumption of sugar sweeteners and potatoes and a high consumption of unsweetened green tea in Japan may be partly related to a lower prevalence of obesity and lower rates of obesity-related diseases [56,64]. A detailed analysis of Japanese dietary patterns, derived from a principal component analysis of the consumption of 134 food items based on 36,737 men and 44,983 women aged 45–74 years, showed that a prudent dietary pattern, with daily food intakes according to the highest quartile (Q4) of dietary pattern scores (which are characterized by high intakes of vegetables (305 g), fruits (285 g), soy products (103 g), potatoes (40 g), seaweed (16 g), mushrooms (16 g), and fish (94 g) and total energy (1824 kcal) consisting of carbohydrates (56.9% energy), proteins (14.6% energy), and fats (25.4% energy)) was significantly associated with a decreased risk of all-cause mortality [65]. In Japan, even in the most Westernized pattern group (Q4), the daily intake of meat was only 73 g compared with that of the prudent group (46 g) [65], compared to 115 g in the U.S. (Figure 9). The longitudinal relationships between different diet indices and dietary patterns with the risk of obesity were investigated in Australia, where dietary habits and the incidence of obesity are close to those in the U.S. [66]. The dietary inflammatory index (DII), plant-based dietary index (PDI), and factor-derived dietary pattern scores have been computed based on food frequency questionnaire data [66]. In the adjusted model, the results of a multivariable log-binomial logistic regression showed that a prudent dietary pattern (RR Q5 vs. Q1 = 0.38; 95% CI: 0.15–0.96) and healthy PDI (RR = 0.31; 95% CI: 0.12–0.77) and overall PDI (RR = 0.56; 95% CI: 0.23–1.33) were inversely associated with obesity risk [66]. Conversely, the DII (RR = 1.59; 95 %CI: 0.72–3.50), a Western dietary pattern (RR = 2.16; 95% CI: 0.76–6.08), and an unhealthy PDI (RR = 1.94; 95%CI: 0.81–4.66) were associated with an increased risk of obesity [66]. The average DII of the Western diet is 1.52 (0.68–3.41) [66], whereas that of the Japanese diet is only 0.82 ± 1.75 [67]. Although an exact comparison of indices of the dietary patterns between Japan and the U.S. is difficult, the Japan public health center-based prospective study [65] and NIPPON DATA2010 [67] showed much lower risks of obesity than in the U.S. In addition to nutrients that affect the degree of obesity and immunity, there are many other dietary components related to COVID-19 infection. These components must also be considered when comparing Japan’s and the U.S.’s DII values, including not only nutrients, but also anti-inflammatory components [66,67] that are important in preventing COVID-19 infection. Low DII and a high consumption of healthy plant-based foods were associated with a lower risk of developing obesity [66]. The Japanese dietary pattern ranks very low in terms of the risk of developing obesity. In fact, the Energy-Adjusted Dietary Inflammatory Index (E-DII) was found to predict the incidence and severity of COVID-19 [68]. Patients with the maximum pro-inflammatory energy-adjusted E-DII score had 7.26-times greater odds of developing COVID-19 compared with those in tertile 1 (E-DII T3 vs. E-DII T1: OR = 7.26; 95% CI: 2.64–9.94, *p* < 0.001) [68]. Furthermore, a positive association was observed between E-DII and C-reactive protein (BE-DII = 1.37; 95% CI: 0.72–2.02), such that with each unit increase in E-DII, the CRP level increased by 1.37 units [68]. A very low COVID-19 mortality in Japan may be attributed to the low DII of the Japanese dietary pattern [67].

## 4. Discussion

### 4.1. Socioeconomic Factors

The large difference in the numbers of patients and deaths due to COVID-19 per capita between Japan and the U.S. cannot be fully explained by socioeconomic factors, because Japan is a highly aged society [4]. Ethnic differences between Japan and the U.S. also cannot fully explain the difference in COVID-19 infection, because the numbers of cases (Figure 1) and deaths (Figure 2) in Asians living in the U.S. are close to those of Whites [2]. Detailed and systematic reviews of racial ethnic disparities in COVID-19-related infections [69], respiratory infections [8], and diabetes mellitus [9] have been published. Health care access and exposure factors may underlie the observed disparities in COVID-19, based on 37 mostly fair-quality cohort and cross-sectional studies and 15 mostly good-quality ecological studies [69]. However, ethnic differences in diet and nutrition were not considered [69], which is why the present study has attempted to elucidate the importance of the role of nutritional differences in preventing COVID-19 infection. The high prevalence (Figure 1) and mortality (Figure 2) of COVID-19 in minority populations is thought to be due to an unhealthy diet [70,71], in addition to poor access to medical care [69]. Survey results from low-income, minority communities showed an estimated average daily intake of salty snacks, candy, cookies, and sugar-sweetened beverages of 532 kcal, which is 88% higher than that recommended in the US Department of Agriculture/Department of Health and Human Services guidelines, and these junk foods were strongly related to the high BMI observed in minority people [70]. In a study of diet quality and risk and severity of COVID-19, COVID-19 cases were documented during over 3,886,274 person-months of follow-up [71]. Here again, a diet characterized by healthy plant-based foods, like the Japanese diet, was associated with a lower risk and severity of COVID-19. This association may be evident among individuals living in areas with higher socioeconomic deprivation [71].

### 4.2. Dietary and Nutritional Factors

A search for “COVID-19 and mortality and obesity and Japanese diet” in PubMed did not find a single paper. However, the Japanese diet is similar to that characterized in a previous study as containing healthy plant-based foods, which was associated with a lower risk and severity of COVID-19 [58,71]. This association may be particularly evident among individuals living in areas of higher socioeconomic deprivation [69]. In fact, a systematic literature review identified nutritional interventions that might prevent or aid in recovery from COVID-19 [3]. A varied and balanced diet, or so-called “nutrient-dense dietary pattern” [56], has low amounts of saturated fatty acids and sugar and an abundance of essential nutrients, such as vitamin D, which are all known to contribute to the normal functions of the immune system [3]. Avoidance of a low DII [66,67,68], excess saturated fats and sugar [56,63,65], and deficiencies [3] and identification of suboptimal intakes of micronutrients in targeted groups of patients in distinct and highly sensitive populations could help to strengthen people’s resilience to COVID-19 [3]. Thus, wider access to healthy foods should be a top priority, and individuals should be mindful of healthy eating habits to reduce their susceptibility to COVID-19 [3,54].

### 4.3. The Japanese American

By comparing Japanese persons living in Japan and Japanese persons living in the U.S., it is possible to evaluate the effects of nutrition and living environment on COVID-19 severity by excluding the effects of genes. The effects of lifestyle westernization on the prevalence of obesity and metabolic diseases have been investigated by comparing Japanese Americans and native Japanese persons [72]. Japanese Americans have the same frequencies of single nucleotide polymorphisms as Japanese persons living in Japan [11,33], but they live in the U.S. environment with corresponding eating and exercise habits. The longitudinal observational data from 765 Japanese Americans over a 10-year period showed that the development of obesity (35.4%) was associated with U.S. dietary habits, such as a high intake of sugar (*p* < 0.001) and fructose (*p* < 0.011) and low intakes of vegetable protein and complex carbohydrates [72]. Furthermore, among obese Japanese Americans, the development of diabetes mellitus was associated with higher intakes of animal protein (*p* = 0.011), animal fat (*p* = 0.010), and saturated fatty acids (*p* = 0.047) [72]. Nutrigenomic studies have shown different genetic backgrounds between American Whites and Japanese people [11,33]. However, in a study of two genetic risk scores for obesity, based on 31 or 68 single nucleotide polymorphisms, no associations were found between the scores and changes in obesity markers, and no significant gene–diet interactions were found [73]. As mentioned previously, obesity is a major risk factor for COVID infection [5,15,74]. In a meta-analysis, 10,233 COVID-19 patients with obesity (33.9%) had higher odds of poor outcomes than better outcomes with a pooled OR of 1.88 (95% CI: 1.25–2.80; *p* = 0.002) [73,74]. Obesity-induced chronic diseases are also risk factors for COVID-19 infection [75]. A total of 28 studies that included 17,000,000 samples (in patients aged 48–80 years) showed that the most prevalent underlying conditions in patients with COVID-19 were hypertension (range: 15–69%), diabetes mellitus (8–40%), cardiovascular disease (4–61%), chronic pulmonary disease (1–33%), and chronic kidney disease (range: 1–48%) [75]. These conditions were each associated with an increased in-hospital case fatality rate ranging from 1% to 56% [75]. In addition, the Japanese diet contains many more functional substances that prevent COVID-19 infection, including fish fatty acids (EPA/DHA) [42,43], soybean isoflavones [45,46,47], seaweed components (fucoidan, porphyrin, fucoxanthin, fucosterol) [48,49], green tea (epigallocatechin-3-gallate) [50,51,52,53], and rice oryzanol and GABA [54,55]. Thus, the nearly 10-times greater prevalence of obesity in the U.S. than in Japan (Figure 3) is induced by the U.S. diet (Figure 8 and Figure 9), and the low incidence and mortality due to COVID-19 in Japan can be partly attributed to the protective effects of the many functional substances in Japanese foods. Therefore, the extremely high incidence of and mortality from COVID-19 in the U.S. compared with Japan are not caused strictly by genetic differences [72,73]. The racial and ethnic disparities in COVID-19-related hospitalizations and deaths in the U.S. are socioeconomic factors [69,74] that cause poor dietary conditions and poor access to medical care.

### 4.4. Immunological Considerations Regarding Diet and Obesity

Obesity is a predictor of the outcome of COVID-19 patients [74], and obesity-induced chronic diseases are also associated with the risk of COVID-19 infection [75]. The role of nutrition in COVID-19 susceptibility and severity has been extensively reviewed [76]. Assuming that nutrition and diet affect the difference in infections and deaths due to COVID-19 between Japan and the U.S., it must be proven that obesity caused by nutrition and diet affects immunity. In fact, it has already been shown that an optimal immune response is dependent on adequate diet and nutrition [3,77]. Frequently, a poor nutrient status is associated with inflammation and oxidative stress, which can, in turn, impact the immune system [3]. There are clear and strong relationships between micronutrients and DHA status and immune function [78]. The prevention of COVID-19 infection by avoiding low-grade chronic inflammation with macro- and micronutrients and bioactive food components has been addressed [79]. Supplements may be effective in preventing COVID-19 infections in people with micronutrient deficiencies [32]. However, high-dose nutrient supplementation is not recommended to prevent COVID-19 infection [76]. A detailed review of 2732 articles from PubMed and EMBASE and 4164 articles from preprint servers, etc. concluded that, currently, there is limited evidence that high-dose supplements of micronutrients either prevent severe disease or hasten recovery [76]. U.S. dietary consumption activates the innate immune system and impairs adaptive immunity, leading to chronic inflammation and an impaired host defense against viruses [77]. The common contributing factors of chronic inflammation were suggested in COVID-19 and inflammatory bowel disease [80]. Obese persons are vulnerable to COVID-19 because obesity-induced expansion of adipocytes alters the function and architecture of adipose tissue. Enlarged adipocytes then become apoptotic and attract macrophages and other cells to form inflammatory adipose tissue [5]. Excessive saturated fatty acid consumption can induce a lipotoxic state and activate the innate immune system via activation of Toll-like receptor 4 (TLR4) expressed on macrophages, dendritic cells, and neutrophils. This triggers the activation of canonical inflammatory signaling pathways that produce pro-inflammatory mediators and other effectors of the innate immune system [81]. The TLR4 signaling pathway is acknowledged to be one of the main triggers of the chronic low-intensity inflammatory response that is induced by obesity [81]. Normal adipose tissue contains three anti-inflammatory cell types (T helper (Th2) cells, M-2 macrophages, and regulatory T-cells (Treg)), which are negative regulators of inflammation [5]. Obesity decreases Th2 cells, Treg cells, and M-2 macrophages. There is an increase in the abundance of pro-inflammatory cells such as CD8+ T cells and M-1 macrophages [5]. Obese, inflamed adipose tissue is composed of 40% M-1 macrophages, which are the source of pro-inflammatory cytokines leading to local as well as systemic inflammation. The ultimate result is a state of chronic inflammation at local and systemic levels [5].

### 4.5. Evidence of a Causal Relationship between COVID-19 Infection and Obesity: Mendelian Randomization Approach

A search of PubMed identified 1209 papers on the associations of COVID-19 infection, obesity, and mortality. However, a correlation does not mean a causal relationship, which must generally be established in randomized, controlled trials. It is ethically, temporally, and technically impossible to induce obesity in subjects via consumption of an American diet and subsequently infect them with COVID-19. Thus, a Mendelian randomization approach has been used to study the causal impact of obesity on the susceptibility and severity of COVID-19 [82]. BMI, waist circumference (WC), and trunk fat ratio (TRF) were measured as indicators of obesity. Summary statistics of genome-wide association studies for these body composition measures were drawn from the GIANT consortium and UK Biobank, and 524 BMI-related, 42 WC-related, and 33 TFR-related independent single nucleotide polymorphisms (SNPs) were considered for Mendelian randomization analyses [82]. As in the U.S., the number of COVID-19 cases and deaths in the United Kingdom were 13.9 and 14.6 times that of Japan on 2 January 2022, respectively [1]. Data from the COVID-19 Host Genetics Initiative were used to analyze susceptibility and severity due to COVID-19 disease [82]. Total and direct causal effect estimates were calculated using SNPs, sensitivity analyses were performed applying Mendelian randomization techniques, and mediation effects of type 2 diabetes mellitus and cardiovascular diseases (CVD) were investigated [82]. A genetically predicted BMI was strongly associated with both susceptibility (OR = 1.31 per 1 SD increase; 95% CI: 1.15–1.50; *p*-value = 7.3 × 10^−5^) and hospitalization (OR = 1.62 per 1 SD increase; 95% CI: 1.33–1.99; *p*-value = 2.8 × 10^−6^) [82]. This result provides strong evidence of a causal impact of overall obesity on the susceptibility and severity of COVID-19 disease [82]. Novel data indicate that obesity and obesity-induced metabolic risk factors might also promote vaccine breakthrough SARS-CoV-2 infections in fully vaccinated people [83]. In both Japan and the United States, where vaccination has been widely practiced, it was found that there was a large difference (>10 times) between the number of people infected with COVID-19 and the mortality rate compared to before the vaccination [1]. The reason for this finding is thought to be the difference in diet between Japan and the U.S., which causes obesity and finally COVID-19 infection, which occurred even after vaccination on 3 January 2022.

### 4.6. Study Limitations

This study has the following limitations.

#### 4.6.1. Comparison of Many Factors Other Than Diet and Nutrition between Japan and the U.S.

There are various factors affecting COVID incidence other than dietary factors. In this report, the other potentially influencing factors, such as socializing, protective behaviors (masks, distance, disinfection), health care, hygienic behaviors (e.g., hand hygiene), physical activities [84], and education [85] were not analyzed. This is because it is extremely difficult to quantify factors such as behavior and education and compare them between Japan and the U.S. Moreover, it is too complicated to analyze many factors simultaneously in one report. Therefore, the author focused on the intake of foods and nutrients that can be accurately quantified, clarified some of the causes of the difference in the frequency of obesity and diabetes in Japan and the U.S., and further explored the difference in the frequency of COVID-19 infection between Japan and the U.S.

#### 4.6.2. Statistical Analyses of the Nutrients and Foods of Japan and the U.S.

As discussed in the Methods section, considering the sample size and the standard deviation compared with the average value, the large differences in the intakes of nutrients and foods [13,14] discussed are statistically significant. More detailed statistical analyses, including score-based patterns assessed by using dietary variety scores of the Japanese diet [86], are not applicable to the American diet. To calculate the contribution of individual nutrient and food intakes to the prevention of COVID-19 infection, data on the intakes of COVID-19 patients and healthy subjects would be required, but these specific data were not obtained. In addition, randomized, controlled trials with patients being infected with COVID-19 given varying amounts of nutrients would be ethically unacceptable. Another limitation is the time lag of the data. Recently, new NHANES data were reported [87], which are valuable nutritional data during the COVID-19 pandemic in the U.S.

## 5. Conclusions

The difference in the number of COVID-19 cases and deaths per million people between Japan and the U.S. of about 12- to 17-times is partly attributable to large differences in the diets of the two countries, perhaps through lowered immune resistance induced by extreme obesity. The U.S. diet, which is high in junk foods and has a high DII, especially in minority groups, may increase vulnerability to COVID-19 infection. The Japanese dietary pattern is close to a “nutrient-dense dietary pattern” and contains many substances that prevent COVID-19 infection. Although there are various factors affecting COVID incidence, which is a very serious worldwide public health problem, this paper focused on the nutritional aspects, considering the possibility that diet influences obesity and COVID-19 infection, without negating the importance of other factors. The findings of this nutritional study will contribute to increasing resistance against COVID-19 and other infectious diseases in the future.

## Figures and Tables

**Figure 1 nutrients-14-00633-f001:**
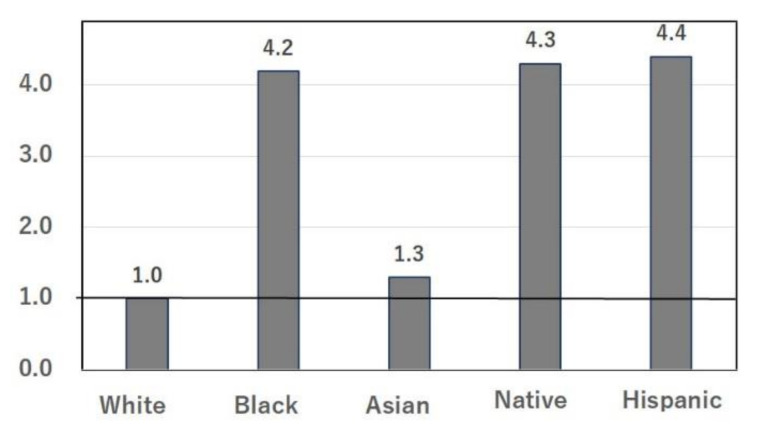
Number of inpatients with COVID-19 according to ethnicity in the U.S. Numbers are relative to the number of Whites as 1. Data were obtained from a cross-sectional study: Centers for Disease Control and Prevention. COVID View. Key updates were for week 43, ending 24 October 2020. Accessed at www.cdc.gov/coronavirus/2019-ncov/covid-data/pdf/covidview-10-30-2020.pdf (accessed 10 December 2021).

**Figure 2 nutrients-14-00633-f002:**
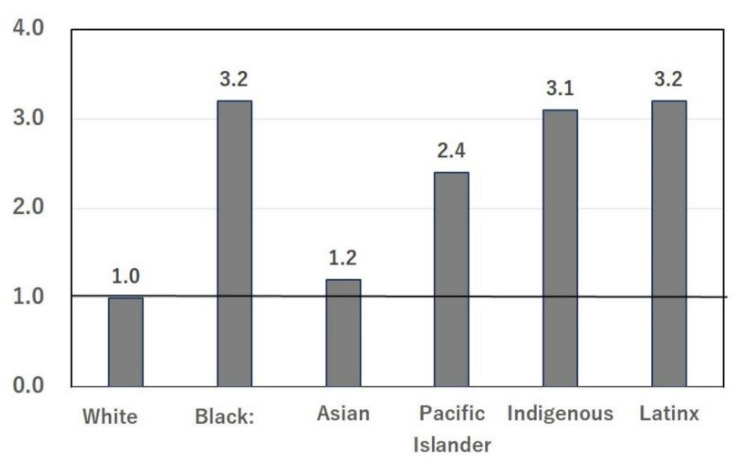
Deaths from COVID-19 according to ethnicity in the U.S. Numbers are relative to the number of Whites as 1. Data were obtained from a cross-sectional study: Centers for Disease Control and Prevention. COVID View. Key updates were for week 43, ending 24 October 2020. Accessed at www.cdc.gov/coronavirus/2019-ncov/covid-data/pdf/covidview-10-30-2020.pdf (accessed 10 December 2021).

**Figure 3 nutrients-14-00633-f003:**
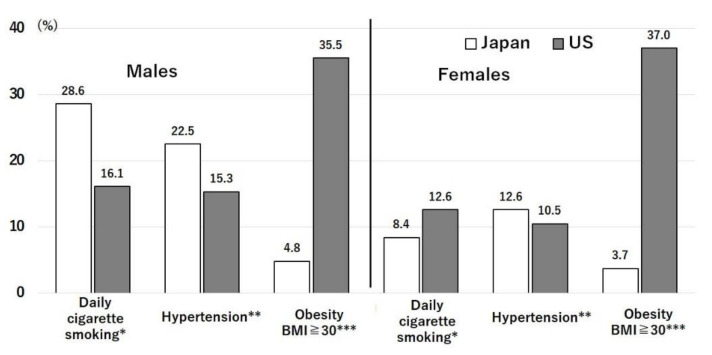
Prevalence of risk factors for non-communicable diseases in Japan and the U.S. Data obtained from the WHO Global Health Observatory data repository, World Health Organization (WHO). Available online: http://apps.who.int/gho/date/note.main.A867?lang₌en (accessed 10 December 2021). * ≥15 years, age-standardized. ** Systolic blood pressure (SBP) ≥ 140 mmHg or diastolic blood pressure (DBP) ≥ 90 mmHg, ≥18 years, age-standardized; *** BMI (body mass index), ≥18 years, age-standardized.

**Figure 4 nutrients-14-00633-f004:**
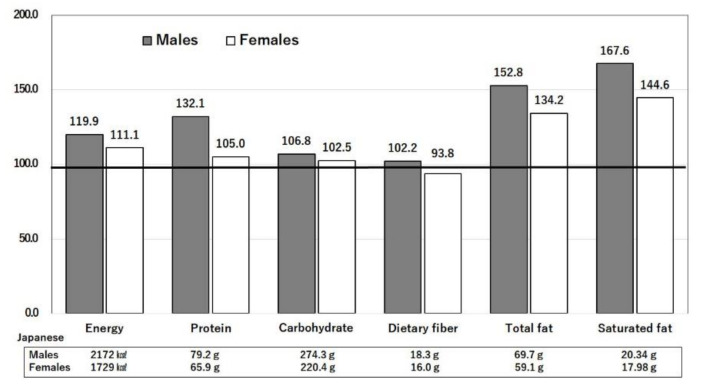
Ratios of macronutrient intakes in the U.S. compared with Japan. Daily intakes of the three major nutrients by Japanese men and women were set to 100% and compared with the intakes of American men and women. The actual daily values of Japanese intakes are shown below each bar. The age range is 40–49 years. United States Department of Agriculture, NHANES 2017–2018, individuals 2 years and over (excluding breast-fed children). www.ars.usda.gov/nea/bhnrc/fsrg (accessed 10 December 2021) and Ministry of Health, Labor and Welfare of Japan. The National Health and Nutrition Survey Japan, 2019. https://www.mhlw.go.jp/bunya/kenkou/kenkou_eiyou_chousa.html (accessed 10 December 2021).

**Figure 5 nutrients-14-00633-f005:**
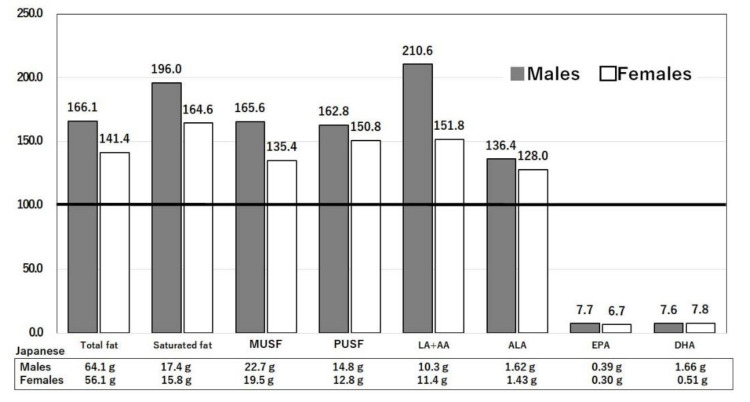
Ratios of lipid intakes in the U.S. compared with Japan. Daily intakes of lipids by Japanese men and women were set to 100% and compared with the intakes of American men and women. The actual daily values of Japanese intakes are shown below each bar. The age range is 40–49 years. AA: Arachidonic acid, ALA: αlinoleic acid, DHA: Docosahexaenoic acid, EPA: Eicosapentaenoic acid, LA: Linoleic acid, MUSF: Monounsaturated fat, PUSF: Polyunsaturated fat. United States Department of Agriculture, NHANES 2017–2018, individuals 2 years and over (excluding breast-fed children). www.ars.usda.gov/nea/bhnrc/fsrg (accessed 10 December 2021) and Ministry of Health, Labor and Welfare of Japan. The National Health and Nutrition Survey Japan, 2019. https://www.mhlw.go.jp/bunya/kenkou/kenkou_eiyou_chousa.html (accessed 10 December 2021).

**Figure 6 nutrients-14-00633-f006:**
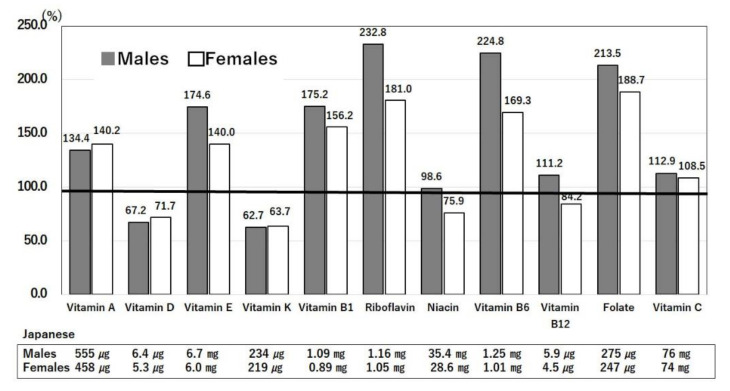
Ratios of vitamin intakes in the U.S. compared with Japan. Intakes of vitamins by Japanese men and women were set to 100% and compared with the intakes by American men and women. The actual values of Japanese intakes are shown below each bar. The age range is 40–49 years. United States Department of Agriculture, NHANES 2017–2018, individuals 2 years and over (excluding breast-fed children). www.ars.usda.gov/nea/bhnrc/fsrg (accessed 10 December 2021) and Ministry of Health, Labor and Welfare of Japan. The National Health and Nutrition Survey Japan, 2019. https://www.mhlw.go.jp/bunya/kenkou/kenkou_eiyou_chousa.html (accessed 10 December 2021).

**Figure 7 nutrients-14-00633-f007:**
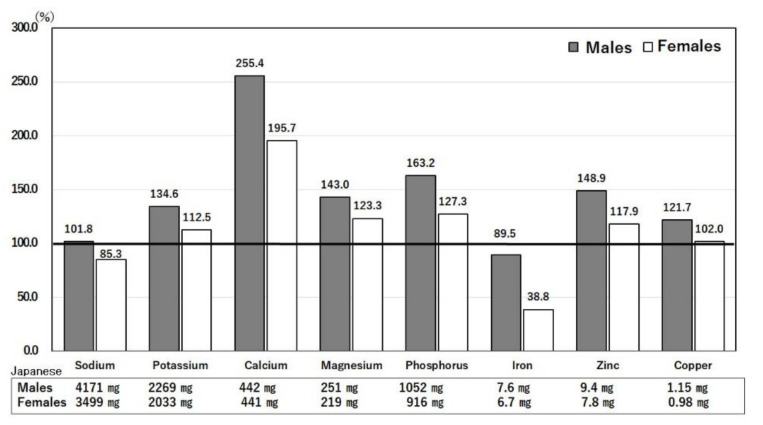
Ratios of mineral intakes in the U.S. compared with Japan. Intakes of minerals by Japanese men and women were set to 100% and compared with the intakes by American men and women. The actual values of Japanese intakes are shown below each bar. The age range is 40–49 years. United States Department of Agriculture, NHANES 2017–2018, individuals 2 years and over (excluding breast-fed children). www.ars.usda.gov/nea/bhnrc/fsrg (accessed 10 December 2021) and Ministry of Health, Labor and Welfare of Japan. The National Health and Nutrition Survey Japan, 2019. https://www.mhlw.go.jp/bunya/kenkou/kenkou_eiyou_chousa.html (accessed 10 December 2021).

**Figure 8 nutrients-14-00633-f008:**
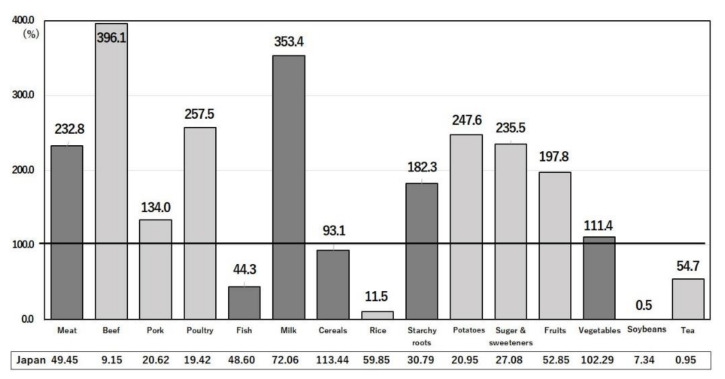
Comparison of food intakes between Japan and the U.S. The value of each item (kg/capita/year) in Japan was taken as 100%. The actual Japanese values are shown below each bar. Data available at http://www.fao.org/faostat/en/#home (accessed 10 December 2021).

**Figure 9 nutrients-14-00633-f009:**
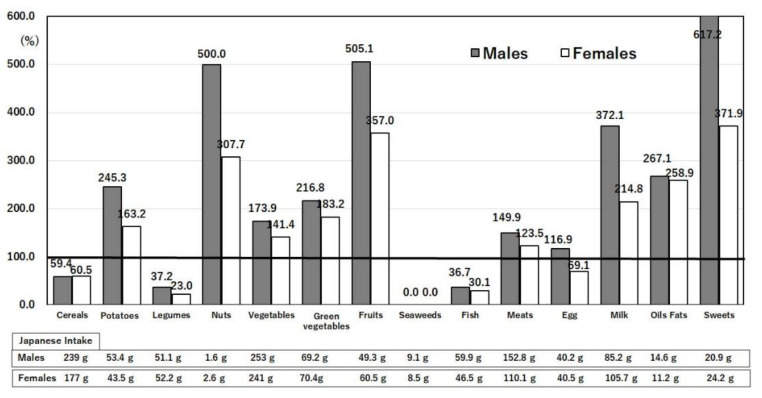
Ratios of food-group intakes in the U.S. compared with Japan. Values are g/day/person in the age range of 40–49 years. The intake of each item in Japan was taken as 100% [14]. The actual values of Japanese intakes obtained from the National Health and Nutrition Survey Japan, 2019, are shown below each bar. United States Department of Agriculture, NHANES 2017–2018, individuals 2 years and over (excluding breast-fed children). www.ars.usda.gov/nea/bhnrc/fsrg (accessed 10 December 2021) and Ministry of Health, Labor and Welfare of Japan. The National Health and Nutrition Survey Japan, 2019. https://www.mhlw.go.jp/bunya/kenkou/kenkou_eiyou_chousa.html (accessed 10 December 2021).

**Figure 10 nutrients-14-00633-f010:**
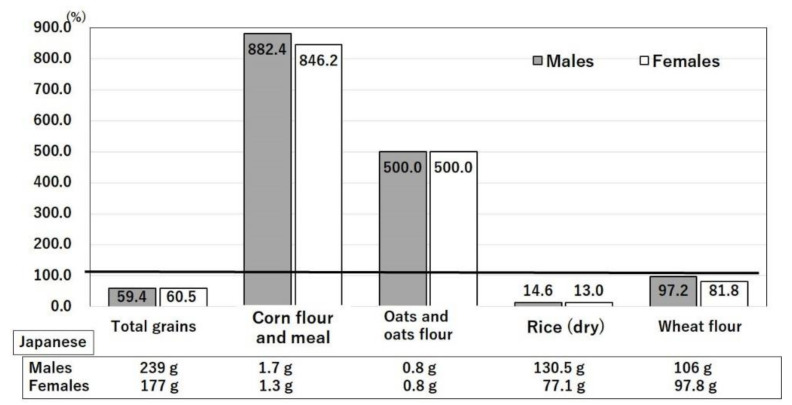
Ratios of grain intakes in the U.S. compared with Japan. Values are g/day/person in the age range of 40–49 years. The intake of each item in Japan was taken as 100% [14]. The actual values of Japanese intakes are shown below each bar. United States Department of Agriculture, NHANES 2017–2018, individuals 2 years and over (excluding breast-fed children). www.ars.usda.gov/nea/bhnrc/fsrg (accessed 10 December 2021) and Ministry of Health, Labor and Welfare of Japan. The National Health and Nutrition Survey Japan, 2019. https://www.mhlw.go.jp/bunya/kenkou/kenkou_eiyou_chousa.html (accessed 10 December 2021).

## Data Availability

No new data were created in this study. Data analyzed in the present study were obtained from the author’s previous publications and online resources.

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
