# Peer review of "Influence of Nutritional Intakes in Japan and the United States on COVID-19 Infection"

_nutrients, 2022, doi:10.3390/nu14030633_

Round 1
Reviewer 1 Report
The manuscript entitled “Influence of Nutritional Intakes in Japan and the United States on COVID-19 Infection” presents interesting issue but some problems must be corrected.
Major:
I have received manuscript which seems to be rather review paper than a regular research study. Author presented some data from the other publications, without any novel data, or any statistical analysis. Author tried to find association between diet that is followed in USA and Japan and (in his opinion resultant) COVID-19 situation in the indicated countries.
There are really important problems associated with the prepared manuscript, including:
- Author totally ignored the other potentially influencing factors, such as socializing, protecting behaviours (masks, distance, disinfection), health care, hygienic behaviours (e.g. hand hygiene) and attributed COVID-19 cases only to the diet which is followed. In spite of the fact that we believe that the diet may influence the COVID-19 cases, we can not state that it is the only influencing factor.
- Author did not analyse any data, but he only presented them. There is no statistical analysis, so Author can not state that there is any influence of the diet.
- The serious flaw of the presented manuscript is associated with the fact, that it presents a highly subjective review, not a systematic review. While the systematic review has a key role for broadening knowledge, the other reviews don’t have such role. Author chosen some studies to present association between diet and COVID-19 cases, but we do not know which studies were chosen and why.
- Taking into account, that the Materials and methods section is not properly presented (it should be broaden), without any specific information, it is hard to understand which studies were included into review and why. Author did not present any key words, which were used during literature search, inclusion and exclusion criteria of references, information about the procedure of literature search conducted by them, number of chosen references, as well as information if some of them were excluded from the review and on the basis of which criteria. As a number of recent publications that are related to the issue were not included, it is a serious problem.
- Author do not present the current and comprehensive knowledge associated with the issue. His study presents misunderstanding and improper approach (he totally ignores various factors influencing the COVID incidence). Such approach that was applied by Author is a very serious ethical problem. Author formulate not justified conclusions for very serious world public health problem. It may cause not only problems associated with scientific misunderstanding, but it may also be life-threatening situation. Author should imagine, that some people may read their review which is not justified, but readers may not know it. The presented information, while applied may cause a problem for a number of COVID patients (based on the presented manuscript, readers may believe that only diet influences COVID-19 incidence).
- Author should reduce number of self-citations.
Author Response
To Reviewer 1
Thank you for your valuable comments and suggestions.
Before providing my point-by-point responses to your comments, I would like to briefly explain why this type of article was written. First, this type of article is strongly needed because it is highly cited. I started to study the nutritional difference between Japan and the U.S. 44 years ago, and I published a paper entitled “Impact of Westernization on the nutrition of Japanese” (Ref. 12), which achieved 342 citations, and the nutritional studies on Japanese islanders (Ref. 19), which also achieved 437 citations in 40 years.
The nutrition surveys including “The National Health and Nutrition Survey, 14. Ministry of Health, Labour and Welfare of Japan” have been performed with the cooperation of Kagawa Nutrition University for many years, as shown in the recent article “The Roles Played by the Institute of Nutrition Sciences, Kagawa Nutrition University on National Nutritional Crises in Japan. Asia Pac J Public Health. 2022 Jan; 34(1):128-130”. Thus, the solid data on Japanese nutrition are partly our original work.
Reviewer 1.
- Author totally ignored the other potentially influencing factors, such as socializing, protecting behaviours…. and attributed COVID-19 cases “only to the diet.”
Ans. As clearly written on lines 20-21 of the Abstract, “Large differences in nutrient intakes and prevalence of diet-induced obesity, but not racial differences, may be partly responsible for differences”.
I agree with you that there are many other potentially related factors, but it is not possible to analyze all these factors, so I concentrated on diet as one of the possible important factors.
- Author did not analyse any data, but he only presented them. There is no statistical analysis, so Author can not state that there is any influence of the diet.
Ans. The 10 figures presented in the manuscript are the comparisons of essential data from Japan and the U.S. This is the essential first starting analysis. The effects of nutrients and foods on obesity and immunity are analyzed by the comparison and references quoted. Based on these, one can state the possibility that diet has an effect on obesity and COVID-19 infection, which was confirmed by Mendelian Randomization.
- The serious flaw of the presented manuscript is associated with the fact, that it presents a highly subjective review, not a systematic review. While the systematic review has a key role for broadening knowledge, the other reviews don’t have such role. Author chosen some studies to present association between diet and COVID-19 cases, but we do not know which studies were chosen and why.
Ans. I chose reliable values on nutritional intakes and health surveys for Figures 1 to 10, which I could not find in other publications in PubMed. Based on the exact comparison between Japan and the U.S., I quoted related references. Since this was mentioned as a “serious flaw”, I have added 21 new relevant references.
- Taking into account, that the Materials and methods section is not properly presented (it should be broaden), without any specific information, it is hard to understand which studies were included into review and why. Author did not present any key words, which were used during literature search, inclusion and exclusion criteria of references, information about the procedure of literature search conducted by them, number of chosen references, as well as information if some of them were excluded from the review and on the basis of which criteria. As a number of recent publications that are related to the issue were not included, it is a serious problem.
Ans. The Materials and Method section was broadened, as suggested (Lines 55-77). As clearly written in my cover letter to the Editors, I used PubMed as the major search engine. A search for "COVID-19 and mortality and obesity" in PubMed found 1,209 papers, and an additional search for "diet" found 39 papers. However, when "Japanese diet" was used as the search term, no papers were found. This was one of the major motivations for writing this paper. In addition, other search engines including CiNii (Japanese database) were also surveyed, using the keywords indicated (Keywords: COVID-19; obesity; saturated fat; EPA/DHA; soybean; diabetes; mortality; and Japanese). The inclusion and exclusion criteria of the references were date after the COVID-19 pandemic in the case of health surveys, and the nutrient and food intakes are the most recent data available (United States Department of Agriculture. NHANES 2017-2018, and the National Health and Nutrition Survey Japan, 2019). Unfortunately, after the COVID-19 pandemic, these nationwide health-nutrition surveys were stopped. These are described in the Materials and Methods section.
Finally, 21 publications were added based on the advice given.
- Author do not present the current and comprehensive knowledge associated with the issue. His study presents misunderstanding and improper approach (he totally ignores various factors influencing the COVID incidence). Such approach that was applied by Author is a very serious ethical problem. Author formulate not justified conclusions for very serious world public health problem. It may cause not only problems associated with scientific misunderstanding, but it may also be life-threatening situation. Author should imagine, that some people may read their review which is not justified, but readers may not know it. The presented information, while applied may cause a problem for a number of COVID patients (based on the presented manuscript, readers may believe that only diet influences COVID-19 incidence). Author should reduce number of self-citations.
Ans. I think it is very clear that various factors affect COVID incidence, which is a very serious worldwide public health problem. However, I focused on the nutritional aspect in this paper and believe that, based on the evidence, one can state the possibility that diet has an effect on obesity and COVID-19 infection, while not excluding other factors (Limitations lines 659-669). This is stated in the final conclusion. Finally, the author’s previous research has been peer reviewed and widely cited and could thus be considered reliable.
English correction
To: Dr. Kagawa
From: Luba W, MD, Forte Inc.
Date: January 19, 2022
Subject: Revisions to revised manuscript (Job No. R2200302)
Reviewer 2 Report
The manuscript,「 Influence of Nutritional Intakes in Japan and the United States on COVID-19 Infection」, is to describe the effects of diet and obesity on preventing COVID-19 infection and conclude that differences in nutrient intakes and prevalence of obesity may be partly responsible for differences in the incidence and mortality of COVID-19 between the U.S. and Japan. Some important weaknesses should be issued clearly.
- Introduction:
1-1. During the COVID-19 outbreak, the numbers of patients and deaths vary widely over time. Comparing the epidemic situation between the two countries may be due to the large differences in the government's policy. Authors should specify the particularity of the time points of outbreak and mortality taken in the introduction section. Alternatively, authors can use morbidity and mortality rates during a period of time to describe how these two countries differ.
1-2. It should more specifically define the hypothesis being tested. - Methods: Methods and protocols should be described in detail and well-established methods can be appropriately cited.
- Results:
3-1. Authors provided some concise and precise description of the results. Figure 1 and 2 show the burden of COVID-19 morbidity and death in U.S. and the results in Figure 1 and 2 are impressive. Why are there no information on the Japanese epidemic in Figure 1 and 2? What would be the result of adding the Japanese ethnic group and comparing the results of Asian in U.S.?
3-2. Figure 3-10 was presented in groups of male and female, whether Figure 1 can also be presented in gender groups, and explain the influence of gender factors in the results and discussion. - Discussion:
4-1. What has Mendelian randomization approach to do with your data?
4-2. It is recommended to have more discussions in the influence of gender factors in diet and COVID-19 infection.
4-3. Research limitations need to be highlighted in the Discussion section.
Author Response
To Reviewer 2
Thank you for your valuable comments and suggestions.
Before providing my point-by-point responses to your comments, I would like to briefly explain why this type of article was written. First, this type of article is strongly needed because it is highly cited. I started to study the nutritional difference between Japan and the U.S. 44 years ago, and I published a paper entitled “Impact of Westernization on the nutrition of Japanese” (Ref. 12), which achieved 342 citations, and the nutritional studies on Japanese islanders (Ref. 19), which also achieved 437 citations in 40 years.
The nutrition surveys including “The National Health and Nutrition Survey, 14. Ministry of Health, Labour and Welfare of Japan” have been performed with the cooperation of Kagawa Nutrition University for many years, as shown in the recent article “The Roles Played by the Institute of Nutrition Sciences, Kagawa Nutrition University on National Nutritional Crises in Japan. Asia Pac J Public Health. 2022 Jan; 34(1):128-130”. Thus, the solid data on Japanese nutrition are partly our original work.
Introduction:
- During the COVID-19 outbreak, the numbers of patients and deaths vary widely over time. Comparing the epidemic situation between the two countries may be due to the large differences in the government's policy. Authors should specify the particularity of the time points of outbreak and mortality taken in the introduction section. Alternatively, authors can use morbidity and mortality rates during a period of time to describe how these two countries differ.
Ans: I compared values before vaccination and after vaccination in the Introduction, as follows. The numbers of patients and deaths per million people due to COVID-19 differ markedly (119,026 and 1,965, respectively, in the U.S. compared with 11,381 and 126, respectively, in Japan, as of 28 August 2021) [1]. Even in the vaccinated, these differences are very large: cases and deaths per million were 167,302 and 2,537, respectively, in the U.S., but only 13,776 and 146, respectively, in Japan, as of 2 January 2022
- It should more specifically define the hypothesis being tested.
Ans. The following sentence was added to the final portion of the Introduction.
I hypothesized that the dietary difference between Japan and the U.S. is one of the major factors explaining the large difference in COVID infection in both countries.
Methods: Methods and protocols should be described in detail and well-established methods can be appropriately cited.
Ans. The following sentences were added to the beginning of this section. Lines 55-77
“PubMed was used as the major search engine. In addition, other search engines including CiNii (Japanese data base) were also used, with the following keywords: COVID-19; obesity; saturated fat; EPA/DHA; soybean; diabetes; mortality; and Japanese.”
A search for "COVID-19 and mortality and obesity" in PubMed found 1,209 papers, and an additional search for "diet" found 39 papers. However, when "Japanese diet" was used as the search term, no paper was found.
The following was added to the last part of this section.
“The inclusion and exclusion criteria of the references were date after the COVID-19 pandemic in the case of health surveys, and the nutrient and food intakes were the most recent data available: United States Department of Agriculture. NHANES 2017-2018 [13] and The National Health and Nutrition Survey Japan, 2019 [14].”
“Unfortunately, after the COVID-19 pandemic, these nationwide health-nutrition surveys were stopped. Reliable data on health, nutrient intakes, and food intakes are compared in Figures 1 to 10. Daily intakes of the nutrients and foods of Japanese men and women were set to 100% and compared with the intakes of American men and women.”
The relevant references are quoted according to each subject.
Results:
3-1. Authors provided some concise and precise description of the results. Figure 1 and 2 show the burden of COVID-19 morbidity and death in U.S. and the results in Figure 1 and 2 are impressive. Why are there no information on the Japanese epidemic in Figure 1 and 2? What would be the result of adding the Japanese ethnic group and comparing the results of Asian in U.S.?
Ans. Japanese in Japan are a homogeneous society, and there are very few blacks and Hispanics.
3-2. Figure 3-10 was presented in groups of male and female, whether Figure 1 can also be presented in gender groups, and explain the influence of gender factors in the results and discussion.
Ans. Unfortunately, the exact gender values were not available.
Discussion:
4-1. What has Mendelian randomization approach to do with your data?
Ans. The randomized control study on long term diet to induce obesity with artificial inoculation of corona vaccine is ethically and technically impossible. Mendelian randomization can overcome these difficulties.
4-2. It is recommended to have more discussions in the influence of gender factors in diet and COVID-19 infection.
Ans. Unfortunately, the exact values by sex were not available.
4-3. Research limitations need to be highlighted in the Discussion section.
Ans. I added the limitations to the Discussion section. Lines 659-669.
English correction:
To: Dr. Kagawa
From: Luba W, MD, Forte Inc.
Date: January 19, 2022
Subject: Revisions to revised manuscript (Job No. R2200302)
Round 2
Reviewer 1 Report
The manuscript entitled “Influence of Nutritional Intakes in Japan and the United States on COVID-19 Infection” presents interesting issue but some problems must be corrected. Unfortunately, the major problems associated with the presented manuscript are not corrected and the manuscript is still misleading.
Major:
I have received manuscript which seems to be rather review paper than a regular research study. Author presented some data from the other publications, without any novel data, or any statistical analysis. Author tried to find association between diet that is followed in USA and Japan and (in his opinion resultant) COVID-19 situation in the indicated countries.
There are really important problems associated with the prepared manuscript, including:
- Author totally ignored the other potentially influencing factors, such as socializing, protecting behaviours (masks, distance, disinfection), health care, hygienic behaviours (e.g. hand hygiene) and attributed COVID-19 cases only to the diet which is followed. In spite of the fact that we believe that the diet may influence the COVID-19 cases, we can not state that it is the only influencing factor.
- Author did not analyse any data, but he only presented them. There is no statistical analysis, so Author can not state that there is any influence of the diet.
- The serious flaw of the presented manuscript is associated with the fact, that it presents a highly subjective review, not a systematic review. While the systematic review has a key role for broadening knowledge, the other reviews don’t have such role. Author chosen some studies to present association between diet and COVID-19 cases, but we do not know which studies were chosen and why.
- Taking into account, that the Materials and methods section is not properly presented (it should be broaden), without any specific information, it is hard to understand which studies were included into review and why. Author did not present any key words, which were used during literature search, inclusion and exclusion criteria of references, information about the procedure of literature search conducted by them. As a number of recent publications that are related to the issue were not included, it is a serious problem.
- Author do not present the current and comprehensive knowledge associated with the issue. His study presents misunderstanding and improper approach (he totally ignores various factors influencing the COVID incidence). Such approach that was applied by Author is a very serious ethical problem. Author formulate not justified conclusions for very serious world public health problem. It may cause not only problems associated with scientific misunderstanding, but it may also be life-threatening situation. Author should imagine, that some people may read their review which is not justified, but readers may not know it. The presented information, while applied may cause a problem for a number of COVID patients (based on the presented manuscript, readers may believe that only diet influences COVID-19 incidence).
- Author should reduce number of self-citations.
Reviewer 2 Report
The authors need to revise the discussion paragraphs. It is not appropriate to extend the discussion if it is not a research result.
